# Achieving Universal Health Coverage (UHC): Dominance analysis across 183 countries highlights importance of strengthening health workforce

**Michael Reid** [1,2] *, **Reena Gupta** [1], **Glenna Roberts** [2], **Eric Goosby** [1,2], **Paul Wesson** [1,3]

**1** School of Medicine, University of California San Francisco, San Francisco, California, United States of America, **2** Institute for Global Health Sciences, University of California San Francisco, San Francisco, California, United States of America, **3** Center for AIDS Prevention Studies, University of California San Francisco, San Francisco, California, United States of America

* Michael.reid2@ucsf.edu

**Data Availability Statement:** The data underlying this study are third-party data which are publicly available from a separate manuscript published by Hogan et al which is cited in this manuscript.

## Abstract

### Background

Despite increasing political will to achieve Universal Health Coverage (UHC), there is a paucity of empiric data describing what health system indicators are useful surrogates of country-level progress towards UHC. We sought to determine what public health interventions were useful tracers of country-level UHC progress.

### Methods

Across 183 countries we evaluated the extent to which 16 service delivery indicators explained variability in the UHC Service Coverage Index, (UHC SCI) a WHO-validated indicator of country-level health coverage. Dominance analyses, stratifying countries by World Bank income criteria, were used to determine which indicators were most important in in predicting UHC SCI scores.

### Findings

Health workforce density ranked first overall, provision of basic sanitation and access to clean water ranked second, and provision of basic antenatal services ranked third. In analysis stratified by World Bank income criteria, health workforce density ranked first in Lower Middle Income-Countries (LMICs) (n = 45) and third in Upper Middle Income-Countries (UMICs) (n = 51).

### Conclusions

While each country will have a different approach to achieving UHC, strengthening the health workforce will need to be a key priority if they are to be successful in achieving UHC.

**Funding:** The authors received no specific funding for this work

**Competing interests:** The authors have declared that no competing interests exist.

## Introduction

Universal Health Coverage (UHC) is a central target of the 2030 Sustainable Development Goals (SDG).[1] While there are numerous proposed strategies for achieving UHC[2] the objectives of UHC are the same, regardless of approach: improving access to health services, improving the health of individuals covered and providing financial risk protection.[3]

Recently, World Health Organization (WHO) and the World Bank proposed the use of a UHC Service Coverage Index (UHC SCI) tool to assess country-coverage with a range of different interventions.[3] This tool summarizes country-level coverage of 16 essential service indices across four core domains; (a) reproductive, maternal, newborn and child health, (b) infectious diseases, (c) non-communicable diseases and (d) service capacity and access, among the general population and the most disadvantaged populations.[3] The UHC SCI is unique insofar as it provides a means for assessing the breadth of essential services being offered by individual countries. Furthermore, it provides a simple and standardized summary of a variety of complex and heterogenous service parameters. While the index is limited by the quality of country-specific data on which it is based and the fact that much of the data has been collected asynchronously,[4] it offers valuable service-oriented insights into the implementation of UHC on a global scale.

In a separate analysis, we investigated whether country-level tuberculosis treatment coverage could serve as a useful surrogate for measuring progress towards UHC.[4] In this brief report, we describe a surprising finding relating to the importance of other indicators in achieving UHC across diverse country settings.

## Methods

The rationale and methodology for how UHC SCI is derived has been described in depth elsewhere.[3] Briefly, the index is constructed from geometric means of 16 tracer indicators which measure: family planning programs, pregnancy and delivery care (measured as four or more visits to antenatal care during pregnancy), child immunization, coverage of pediatric services, HIV treatment services, TB treatment coverage, malaria prevention interventions, basic and water sanitation services, cardiovascular disease treatment coverage, cancer and diabetes screening capability, tobacco control measures, access to hospital services, health care worker density (measured as the number of physicians, surgeons and psychiatrists per person in each country), health security (measured based on International Health regulations core capacity index) and an indicator of access to essential medicines. Each indicator is calculated based on publicly reported data and a standardized approach applied across all countries. For each country, all 16 indices were aggregated and reported as a score, measured on a scale of 0–100%, with 100% representing achievement of universal health coverage.

Notably, the hospital bed density and health worker density indicators have a lower bound of 0 but do not have a clear optimal level of maximum. For these variables, a threshold value was selected based on observed minimum values across high-income Organization for Economic Co-operation and Development (OECD) countries. Countries with values above the thresholds for these indicators where held at 100 and those below were linearly rescaled between 0 and 100.

To determine the relative importance of each indicator to the overall UHC SCI score we performed a 'dominance analysis', using the epsilon methodology.[5] This statistical approach relies on estimating the $R^2$ values of all possible combinations of predictors and measures the relative importance by doing pairwise comparisons of all predictors in the model as they relate to an outcome variable, such as the composite score evaluated in this analysis. [6] The relative importance of each indicator was determined by the size of the effect without any inference on

the relative 'importance' of the other variables. Dominance analysis is well suited to answering questions of 'importance' among a set of variables such as this, especially since multi-collinearity would undermine use of traditional multiple regression approaches.[7] Dominance analysis was performed for all countries, collectively, and then separately for countries within strata of World Bank income categories; high income countries (HIC), upper middle-income countries (UMICs), lower middle-income countries (LMICs) and low-income countries (LICs). Indicators for malaria, cancer screening coverage, and essential medications, were not included in this analysis due to lack of consistent reporting for the majority of countries. All analyses were conducted using Stata version 14 (College Station, TX).

## Results

Across 183 countries, the median service coverage index was 65 (Interquartile range [IQR]: 48, 75), with 44 countries having UHC SCI scores greater than 75 of which 35 were classified as HICs. There were 45 countries in the lowest quartile, of which 40 were classified as LICs.

The results of the dominance analysis assessing all 183 countries, collectively, indicated the most dominant interventions were density of the health workforce (ranked first), provision of basic and water sanitation services (ranked second) and provision of pregnancy and delivery care (ranked third) (Table 1). When stratified by World Bank income categories, dominance analysis ranked health workforce density fifth among LICs (n-34), first among LMICs (n = 45), third among UMICs (n = 51) and eleventh among HICs (n = 53). Across World Bank income categories, access to hospital services ranked first in LICs, family planning first in UMICs and provision of HIV treatment first in HICs.

## Discussion

In this analysis, health workforce density ranked as a critical element in determining UHC service coverage in LICs, LMICs and UMICs. Improving health service coverage in these countries depends on the availability, accessibility, and capacity of health workers to deliver high quality people-centered integrated care. While other dimensions of service provision are also important, as our dominance analysis highlights, investment must focus on investing in developing the health workforce. To meet the health workforce requirements of the Sustainable Development Goals and UHC targets outlined in the UNHLM, over 18 million additional health workers are needed by 2030.[8] The growing demand for health workers is projected to add an estimated 40 million health sector jobs to the global economy by 2030.[9] Our analysis underscores the importance of investments from both public and private sectors in health worker education, as well as in the creation and filling of funded positions for health care workers once they complete pre-service training. Optimally aligning HRH investments and developing targeted strategies to ensure UHC demands a thorough understanding of unique, country-specific labor dynamics. Policies need to take into account determinants of both the supply and the demand for health workers, how these interact and how this interaction varies in different countries.[10] While many countries have taken substantial steps to increase the numbers of skilled health workers in recent years, less attention is paid to the quality and efficiency of the health workforce, and the resulting requirements of the health system.[11] A holistic approach to improving the health workforce also demands greater attention to enhanced training and adequate mentoring; use of evidence to inform interprofessional education; a shift from predominantly hospital-based care to care in the community; and increased attention to the importance of robust management capacity, especially in low and middle income countries. [12]

**Table 1.  Dominance analysis rankings service coverage indicators based as predictors of UHC SCI score, stratified by World Bank country ranking.**

| | All countries (n = 183) | | LICs (n = 34) | | LMICs (n = 45) | | UMICs (n = 51) | | HICs (n = 53) | |
|---|---|---|---|---|---|---|---|---|---|---|
| | Std. Dominance Coefficient | Rank | Std. Dominance Coefficient | Rank | Std. Dominance Coefficient | Rank | Std. Dominance Coefficient | Rank | Std. Dominance Coefficient | Rank |
| **Family planning[a]** | 0.09 | 4 | 0.084 | 6 | 0.12 | 3 | 0.15 | 1 | 0.10 | 4 |
| **Pregnancy and delivery care[b]** | 0.1 | 3 | 0.05 | 9 | 0.09 | 4 | 0.04 | 10 | 0.08 | 5 |
| **Child immunization[c]** | 0.06 | 11 | 0.08 | 7 | 0.04 | 9 | 0.08 | 7 | 0.01 | 13 |
| **Child treatment[d]** | 0.08 | 7 | 0.05 | 11 | 0.06 | 7 | 0.3 | 11 | 0.04 | 9 |
| **HIV treatment[e]** | 0.06 | 10 | 0.01 | 12 | 0.04 | 10 | 0.11 | 4 | 0.21 | 1 |
| **Tuberculosis effective treatment[f]** | 0.06 | 9 | 0.11 | 3 | 0.08 | 5 | 0.01 | 5 | 0.06 | 7 |
| **Water and sanitation[g]** | 0.12 | 2 | 0.14 | 2 | 0.16 | 2 | 0.05 | 9 | 0.03 | 12 |
| **Prevention of cardiovascular disease[h]** | 0.07 | 8 | 0.012 | 4 | 0.03 | 12 | 0.09 | 6 | 0.14 | 2 |
| **Management of diabetes[i]** | 0.01 | 13 | 0.01 | 13 | 0.02 | 13 | 0.02 | 12 | 0.07 | 6 |
| **Tobacco control[j]** | 0.05 | 12 | 0.05 | 10 | 0.03 | 11 | 0.07 | 8 | 0.12 | 3 |
| **Hospital access[k]** | 0.09 | 5 | 0.15 | 1 | 0.05 | 8 | 0.01 | 13 | 0.03 | 10 |
| **Health care worker density[l]** | 0.13 | 1 | 0.11 | 5 | 0.2 | 1 | 0.13 | 3 | 0.03 | 11 |
| **Health security[m]** | 0.09 | 6 | 0.06 | 8 | 0.08 | 6 | 0.15 | 2 | 0.06 | 8 |

[a] Family planning–measured as demand satisfied with modern methods in women aged 15–49 years who are married or in a union (%)

[b] Pregnancy and delivery care–measured as four or more visits to antenatal care (%)

[c] Child immunization–measured as children aged 1 year who have received three doses of diphtheria, tetanus and pertussis vaccine (%)

[d] Child treatment–measured as care-seeking behavior for children with suspected pneumonia (%)

[e] HIV treatment–measured as the proportion of people with HIV receiving antiretroviral treatment (5)

[f] Tuberculosis treatment–measured as TB effective treatment, calculated as ration of rate of case detection to rate of TB treatment

[g] Water and sanitation—measured as the proportion of households with access to at least basic sanitation (%)

[h] Prevention of cardiovascular disease–measured as prevalence of non-raised blood pressure regardless of treatment status (%)

[i] Management of diabetes–measured as mean fasting plasma glucose measured in country-specific household surveys

[j] Tobacco control–measured as adults aged 15 years or older who had not smoked tobacco in the previous 30 days (%)

[k] Hospital access—measured as the number of hospital beds per person in each country

[l] Health care worker density—measured as the number of health professionals per person, comprising physicians, psychiatrists and surgeons

[m] Health Security–measured based on International Health regulations core capacity index. Malaria prevention is also included in the UHC SCI for countries where malaria is prevalent. Since most countries do not collect this data, we excluded it from our analysis. Cervical cancer screening and access to essential medicines were excluded because of low data availability.

**Abbreviations:** HIC—High-Income Countries, UMICs—Upper Middle-Income Countries, MICs- Middle-Income Countries, LMICs—Lower Middle-Income countries, LICs—Low-Income Countries.

In addition to highlighting HRH issues as a key element to ensuring UHC, our analysis also offers insights into the broader set of service priorities which countries should address so as to achieve universal access. Recognizing that any investment in health programs is essentially a *political* rather than a technical choice,[13] we assert that this dominance analysis can help inform the political economy around where and how resources are invested in the health system. In LICs and LMICs, our results underscore the importance of access to clean water and basic sanitation, and maternal and reproductive care. Absent these essential elements, it is very difficult for health policymakers and practicing healthcare workers to build a functioning system or implement change effectively. [14]

In HICs, where health workforce density and water sanitation are less important predictors of UHC SCI, HIV and cardiovascular disease programs were the most dominant interventions, highlighting the importance of disease-specific programs in settings where accesses to basic public health provisions, including availability of doctors and access to water and sanitation have often been universally secured. While HICs are able to provide a wide array of health services, most low and middle income countries only have the resources to deliver a smaller set of services, necessitating a more explicit and systematic approach to priority setting.[15] As such these findings also underscore the importance of effective and integrated public health care and preventions programs, and basic hospital access as critical elements to achieving UHC.

We note that direct hospital access is not a high-ranking parameter in HICs, but is of high importance in LICs. We posit that the importance of this parameter in LICs underscores how access to hospitals is a useful parameter for distinguishing those lower income countries that are likely to have other basic health infrastructure and those that are not. Nonetheless, the contradistinction between the importance of this variable in LICs compared to HICs may reflect the fact that this indicator is more accurate in LICs and LMICs, while the number of inpatient hospital admissions is often better documented as a surrogate for hospital availability in HICs. [3] Similarly, the fact that the family planning indicator ranks mostly highly in UMICs, but appears to be of less importance in LICs and MICs, may reflect the fact that this variable, which is calculated using a complex denominator derived from multiple survey questions, is better reported in those UMICs. As with other sub-indices in the SCI, the method for deriving this variable is highly sensitive to the methods and quality of data acquisition across countries. [3] Tracking UHC SCI scores over time, and minimizing missing data across countries would enable further exploration of the relative importance of these different variables and these discrepant findings.

This analysis has a number of important limitations. It provides no sub-national granularity and offers no insights into how human resource shortages disproportionally impacts marginalized populations,[16] even in high-income countries. In addition, the HRH sub-index does not provide any insights into the impact of training or supervision on how the workforce functions.[17] The importance of training infrastructure to ensure that health workers can respond to the changing health needs of the population while also maintaining the quality of services is vital, but not captured in the SCI. Furthermore the HRH sub-index used in calculating the UHC SCI score is based on the density of physicians per 100,000 population. It does not draw attention to differences in other cadres within the health workforce or their importance to achieving UHC. In many countries, basic information fields, such as health worker stock and distribution, are largely limited to physicians, nurses and midwifes, despite the growing role played by other cadres, such as community health workers.[8] Significant improvements in the capacity of countries to understand the conditions and opportunities to strengthen their national labor markets is essential. Crucially, this evidence should be developed through putting in place country-level mechanisms to collate, analyze, and use data on a routine basis, especially for the purpose of tracking overall progress towards UHC. Finally, another key limitation of our analysis relates to the nature of the UHC SCI, and the fact that it does not capture longitudinal data about the dynamics of health system evolution. In addition, neither the HRH sub-index nor the overall UHC SCI score capture other dimensions of UHC related to financing of health services or the quality of service provided. Metrics that track these will continue to be essential as countries plan towards UHC.[15]

In summary, this analysis represents the first attempt to define on an empirical basis the importance of health workforce requirements to UHC progress at a county level. Greater efforts to prioritize and strengthen human resources for health are necessary if countries are going to be successful in realizing the goal of UHC.

Dominance analysis to demonstrate importance of health workforce to achieving UHC

## Author Contributions

**Conceptualization:** Michael Reid, Reena Gupta, Eric Goosby, Paul Wesson.

**Formal analysis:** Michael Reid, Glenna Roberts.

**Methodology:** Paul Wesson.

**Writing – original draft:** Michael Reid.

**Writing – review & editing:** Reena Gupta, Eric Goosby, Paul Wesson.

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
