## [Decision Letter · Decision Letter 0]

6 Jan 2020

PONE-D-19-26015

Achieving Universal Health Coverage (UHC): dominance analysis across 183 countries highlights importance of strengthening health workforce

PLOS ONE

Dear Dr. Reid,

Thank you for submitting your manuscript to PLOS ONE. After careful consideration, we feel that it has merit but does not fully meet PLOS ONE’s publication criteria as it currently stands. Therefore, we invite you to submit a revised version of the manuscript that addresses the points raised during the review process.

This is a well written short manuscript which will further be strengthened if it has a strong discussion section. I agree that human resources for health are the prime focus of attention in LMICs to achieve UHC. However, there are other important facets of UHC like quality of care, essential services offered, access to the services and financial protection to avail the services. These facets, though not  apart of the analysis, requires to be touched upon and discussed. Apart from this, I think the reviewer has also raised some very important points which needs to be addressed in the manuscript. 

We would appreciate receiving your revised manuscript by 15th January 2020. Please include the following items when submitting your revised manuscript:

We look forward to receiving your revised manuscript.

Kind regards,

Manu Raj Mathur, PhD

Academic Editor

PLOS ONE

Journal Requirements:

-Hogan, Daniel R., et al. "Monitoring universal health coverage within the Sustainable Development Goals: development and baseline data for an index of essential health services." The Lancet Global Health 6.2 (2018): e152-e168.

https://adra.eu/Health-For-All.pdf

In your revision ensure you cite all your sources, and quote or rephrase any duplicated text outside the methods section. Further consideration is dependent on these concerns being addressed.

Additional Editor Comments (if provided):

This is a well written short manuscript which will further be strengthened if it has a strong discussion section. I agree that human resources for health are the prime focus of attention in LMICs to achieve UHC. However, there are other important facets of UHC like quality of care, essential services offered, access to the services and financial protection to avail the services. These facets, though not  apart of the analysis, requires to be touched upon and discussed.

Reviewers' comments:

Reviewer's Responses to Questions

**Comments to the Author**

1. Is the manuscript technically sound, and do the data support the conclusions?

Reviewer #1: Yes

2. Has the statistical analysis been performed appropriately and rigorously? 

Reviewer #1: No

3. Have the authors made all data underlying the findings in their manuscript fully available?

Reviewer #1: Yes

4. Is the manuscript presented in an intelligible fashion and written in standard English?

Reviewer #1: Yes

5. Review Comments to the Author

Reviewer #1: This is a quick and elegant analysis that offers insight for understanding progress on the UHC path. However, it seemed a bit rushed to me and thus lacks depth. This is of course, fairly easily remedied - I have suggestions on this below:

For one, authors have already engaged with the strengths and weaknesses of the UHC index. Highlighting this more is desirable.

Some mention of whether an indicator centric approach to gauging progress should also be discussed. There is obviously more to this, as work on political economy by Bump, Reich, etc. have outlined and the paper is incomplete without mention of these features.

Relevant applications of DA should be referenced and selection of estimation method - least squares was used? or were adjusted methods relied on. Some sensitivity analysis would be appropriate even as conclusions tend to invariant across methods.

The discussion places emphasis on density (even as the limitations of this are later outlined) - some mention of supportive supervision and continuous training - will be especially important as a number of particularly LMIC countries transition their systems from MCH/MDG focused systems and measurement to CPHC including NCDs.

Worthwhile to discuss hospital access in LICs and family planning in UMICs? Place this in some context too.

also, what are the instrumentation and analytical reasons why other factors are lower priority. Despite the fact that the number of LMICs (is fewer in number, for eg than HICs) pregnancy care ranks higher - what is this showing us?

This method is highly sensitive to the quality of data and operationalisation of indicators, which is a big issue. This should be mentioned. There is growing work on the relevance and measurement of these indicators at the ground level that authors may also consider referencing

6. PLOS authors have the option to publish the peer review history of their article (what does this mean?). If published, this will include your full peer review and any attached files.

Reviewer #1: Yes: Devaki Nambiar

---

## [Author Response · Author response to Decision Letter 0]

23 Jan 2020

Response to Reviewer #1

(Reviewer comments are in bold italics)

Comment 1: This is a quick and elegant analysis that offers insight for understanding progress on the UHC path. However, it seemed a bit rushed to me and thus lacks depth. 

Response: We are grateful for the reviewer’s statement that this is an elegant analysis and agree that it does provide valuable insights that can inform UHC policies in countries that intend to expand access to health services. As we outline below, we have expanded the discussion section highlighting in more depth the critical role of human resources for health to the broader UHC agenda. In addition, we have added more content on both the strengths and weakness of dominance analysis.

Comment 2: Authors have already engaged with the strengths and weaknesses of the UHC index. Highlighting this more is desirable.

Response: In the revised draft we have included more detail on the UHC index. In addition, we have included a reference to a more detailed description of the index in another of our papers on this topic. This additional context is on line 63:

‘The UHC SCI is unique insofar as it provides a means for assessing the breadth of essential services being offered by individual countries. Furthermore, it provides a simple and standardized summary of a variety of complex and heterogenous service parameters. While the index is limited by the quality of country-specific data on which it is based and the fact that much of the data has been collected asynchronously, it offers valuable service-oriented insights into the implementation of UHC on a global scale.’

In addition to this edit, we have added statements in the limitation section of the discussion, expanding on why the UHC SCI offers an incomplete picture of universal health coverage and how it offers no insights into subnational disparities in workforce density. On line 220, we state:

‘This analysis has a number of important limitations. The analysis provides no sub-national granularity and offers no insights into how human resource shortages disproportionally influence marginalized populations, even in high-income countries.’

On line 239, we have added the following text:

‘Finally, another key limitation of our analysis relates to the nature of the UHC SCI, and the fact that it does not capture longitudinal data about the dynamics of health system evolution. In addition, neither the HRH sub-index nor the overall UHC SCI score capture other dimensions of UHC related to financing of health services or the quality of service provided. Metrics that track these will continue to be essential as countries plan towards UHC.’

Comment 3: Some mention of whether an indicator centric approach to gauging progress should also be discussed. There is obviously more to this, as work on political economy by Bump, Reich, etc. have outlined and the paper is incomplete without mention of these features.

Response: This comment is helpful and insightful. We agree that the political economy of universal health coverage has a critical bearing on both the design and implementation of efforts to improve health system performance. Furthermore, we acknowledge that using indicators, such as those delineated within the UHC SCI is only one mechanism for determining how each country navigates a path towards UHC. In the discussion section of the paper we highlight how UHC also entails ensuring quality of service and providing mechanisms to fund healthcare, neither of which are addressed in the UHC SCI (line 242).

The purpose of our analysis is not to provide a comprehensive mapping of all the potential political economy factors and strategies related to UHC, but we do provide a structured way for thinking about the relative importance of different essential services that we hope will assist policy-makers in prioritizing what services and resources to invest in. We do not subscribe to the perspective that decisions about UHC policy should only be determined by indicators. Rather we primarily want to highlight in this manuscript how service coverage priorities differ by countries, and that HRH appears to be essential to advancing a UHC agenda in all settings, regardless of country income status. We recognize the essential role of political economy factors throughout the policy cycle, including agenda-setting, policy design, adoption, implementation and evaluation in order to achieve UHC, and have highlighted how an analysis such as ours can help inform this process. On line 181, we have added the following text:

“Recognizing that any investment in health programs is essentially a political rather than a technical choice we assert that this dominancy analysis can help inform the political economy around where and how resources are invested in the health system.”

Comment 4: Relevant applications of DA should be referenced and selection of estimation method - least squares was used? or were adjusted methods relied on. Some sensitivity analysis would be appropriate even as conclusions tend to invariant across methods.

Response: In the revised draft we have included reference to similar and relevant applications of Dominance Analysis, including citations from Azen et al and Sauceda et al. Notably, we performed the Dominance Analysis using the ‘epsilon’ approach in STATA; as proposed by the reviewer we have indicated this in the revised draft (line 95). This approach has been described in depth by Jeff Johnson in Multivariate Behavioural Research (Johnson, J. W. A heuristic method for estimating the relative weight of predictor variables in multiple regression. Multivariate Behavioral Research,35(1), 1-19.) and employs a methodology similar to least squares as part of the approach, utilizing a relative weight analysis to estimate the proportionate contribution to R2 for a set of predictors.

We did explore using other analystic tools to undertake an additional sensitivity analysis in order to determine the variable importance of multiple independent variables in accounting for variance in a single variable. In particular, we explored using variable clustering analysis, which would have enabled us to cluster countries according to which sub-indices from within the UHC SCI had maximal impact, using the beta weights to assess variable importance. We opted against using variable clustering analysis because such an analytic approach would have offered no mechanism for differentiating between relevant and irrelevant variables. In addition, we note that beta weights are also limited in their ability to determine suppression in a regression equation. 

Comment 5: The discussion places emphasis on density (even as the limitations of this are later outlined) - some mention of supportive supervision and continuous training - will be especially important as a number of particularly LMIC countries transition their systems from MCH/MDG focused systems and measurement to CPHC including NCDs.

Response: We agree with the reviewer’s comment. We have included an additional clause in the limitations section of the manuscript, highlighting the importance of training and supervision as crucial elements of a well-functioning health workforce. This additional text is included from line 220:

‘The analysis provides no sub-national granularity and offers no insights into how human resource shortages disproportionally influence marginalized populations, even in high-income countries. In addition, the HRH sub-index does not provide any insights into the impact of training or supervision on how the workforce functions. The importance of training infrastructure to ensure that health workers can respond to the changing health needs of the population while also maintaining the quality of services is vital, but not captured in the SCI.’

We have also included in the discussion the importance health workforce training and mentorship to ensure workforce quality in addition increasing workforce density. On Line 175, we have added the following text:

“While many countries have taken substantial steps to increase the numbers of skilled health workers in recent years, less attention is paid to the quality and efficiency of the health workforce, and the resulting requirements of the health system.[11] A holistic approach to improving the health workforce also demands greater attention to enhanced training and adequate mentoring; use of evidence to inform interprofessional education; a shift from predominantly hospital-based care to care in the community; and increased attention to the importance of robust management capacity, especially in low and middle income countries

Comment 6: Worthwhile to discuss hospital access in LICs and family planning in UMICs? Place this in some context too.

Response: We agree that these are other parameters are worthy of discussion since they appear to have an important role in specific circumstance. To this ends we have added additional sentences highlighting the import of these findings in the discussion. These additional edits begin on line 236.

As outlined in these added sentences, direct hospital access is a fundamental component of healthcare particularly in low income countries. Furthermore, access to hospitals is also a useful parameter for distinguishing those countries that are likely to have other basic health system infrastructure and those poorest countries that are not. Hospital access is a proxy for essential inpatient services and has more data available in low-income and middle-income countries than number of inpatient hospital admissions, which is often better characterized in high-income settings. In calculating the UHC SCI a threshold value is used to capture only low capacity levels because high values might represent overcapacity or inefficient allocation of resources.

Access to family planning services is also a fundamental component of the health system. However, as discussed in the Hogan paper, the index has a relatively complex denominator derived from multiple survey questions and data collection. As a consequence, the indicator is probably better characterized in high income countries, than low-income and middle-income countries, a factor underscored by fact that the data is missing for >60 countries.

These edits are discussed starting on line 206 in the revised draft:

‘We note that direct hospital access is not a high-ranking parameter in HICs, but is of high importance in LICs. We posit that the importance of this parameter in LICs underscores how access to hospitals is a useful parameter for distinguishing those lower income countries that are likely to have other basic health infrastructure and those that are not. Nonetheless, the contradistinction between the importance of this variable in LICs compared to HICs may reflect the fact that this indicator is more accurately in LICs and LMICs, while the number of inpatient hospital admissions is often better documented as a surrogate for hospital availability in HICs. Similarly, the fact that the family planning indicator ranks mostly highly in UMICs, but appears to be of less importance in LICs and MICs, may reflect the fact that this variable, which is calculated using a complex denominator derived from multiple survey questions, is better reported in those UMICs. Tracking UHC SCI scores over time, and minimizing missing data across countries would enable further exploration of the relative importance of these different variables and these discrepant findings.’

Comment 7: What are the instrumentation and analytical reasons why other factors are lower priority. Despite the fact that the number of LMICs (is fewer in number, for eg than HICs) pregnancy care ranks higher - what is this showing us?

Response: This is a great question. The fact that pregnancy care ranks higher in LMICs compared to UMICs perhaps highlights variability in the quality of data across countries. This index measures whether women have access to four or more antenatal care visits during pregnancy and as such captures the amount of contact with the health system but not the quality of care received during pregnancy. Skilled attendance at birth is a preferred alternative for assessing pregnancy services; however insufficient standardized measurement of skilled health-care personnel makes cross-country comparisons difficult. We therefore speculate that the disparity between where this parameter ranks in lower-middle income and upper-middle countries may result because the index ceases to be as sensitive indicator of pregnancy care in settings when comparing countries that are all more likely to have such services, even if the quality of service is variable. 

Comment 8: This method is highly sensitive to the quality of data and operationalisation of indicators, which is a big issue. This should be mentioned. There is growing work on the relevance and measurement of these indicators at the ground level that authors may also consider referencing

Response: We absolutely agree with this comment and have added a subclause on line 215 highlighting this challenge.

---

## [Editor Report · Decision Letter 1]

12 Feb 2020

Achieving Universal Health Coverage (UHC): dominance analysis across 183 countries highlights importance of strengthening health workforce

PONE-D-19-26015R1

Dear Dr. Reid,

We are pleased to inform you that your manuscript has been judged scientifically suitable for publication and will be formally accepted for publication once it complies with all outstanding technical requirements.

With kind regards,

Manu Raj Mathur, PhD

Academic Editor

PLOS ONE

---

## [Editor Report · Acceptance letter]

20 Feb 2020

PONE-D-19-26015R1 

Achieving Universal Health Coverage (UHC): dominance analysis across 183 countries highlights importance of strengthening health workforce 

Dear Dr. Reid:

I am pleased to inform you that your manuscript has been deemed suitable for publication in PLOS ONE. Congratulations! Your manuscript is now with our production department. 

With kind regards,

on behalf of

Dr. Manu Raj Mathur 

Academic Editor

PLOS ONE